# Luminex MFI—Efforts from a Qualitative to a Quantitative Analysis

**DOI:** 10.3390/biology14060686

**Published:** 2025-06-12

**Authors:** Thomas Schacker, Ramona Landgraf, Ilias Doxiadis, Henry Loeffler-Wirth, Claudia Lehmann

**Affiliations:** 1Interdisciplinary Centre for Bioinformatics, IZBI, Leipzig University, 04107 Leipzig, Germany; tommy@schacker.net (T.S.);; 2Laboratory for Transplantation Immunology, Institute for Transfusion Medicine, University Hospital Leipzig, 04103 Leipzig, Germany; 3HLA Laboratory, Institute for Transfusion Medicine, University Hospital Hamburg-Eppendorf, 20246 Hamburg, Germany

**Keywords:** HLA antibodies, Luminex, bead-based solid phase, MFI, ratio, bead variability

## Abstract

Antibody screening is crucial for preparing patients for organ transplantation and for monitoring them afterwards. A common tool for this is the Luminex single-antigen bead assay. The method uses polystyrene beads that are coated with human leukocyte antigens. Incubating them with the patient’s blood allows antibodies to bind to the beads. Adding secondary antibodies with a fluorescent label allows for the detection of a fluorescent signal that correlates with the relative amount of antibodies bound to a bead. Although the test is widely used, it produces inconsistent results due to technical variability. In this study, we explore the variability of this assay by analyzing multiple measurements from one laboratory. We demonstrate that raw MFI values fluctuate too much to reliably reflect true changes in antibody levels. To address this, we propose using a ratio value by testing two samples from the same patient alongside each other. One sample serves as a reference, the other as the current test sample. Comparing them as a ratio can help to cancel out technical noise and highlight real changes in antibody levels. This approach could make test results more reliable and help in making better clinical decisions before and after a transplant.

## 1. Introduction

Since the introduction of the humoral theory of transplantation [1] and the availability of the Luminex-based assay for the detection of HLA-specific alloantibodies, experts in clinical transplantation immunology have discussed how to make use of the results obtained by the Luminex-based assay [2,3]. It became obvious that day-to-day variation and user-to-user variability hamper the results [4,5]. Pre- and post-transplantation management requires solid data for the decision-making process. Interpretation of a decrease or increase in serum antibody levels should therefore be based on quantitative data and not on relative qualitative results. The mean fluorescence intensity (MFI) value is used in most cases to judge antibody abundances in a patient. However, MFI values show an inherent variability of the system and have many factors influencing it, so it is questionable as to whether reporting only the MFI is the best practice [6,7]. MFI values are relative values that are not directly suitable for quantitative evaluation on an absolute scale [6,8]. However, everyday clinical practice regularly requires a quantitative statement in post-monitoring after organ transplantation and for the observation and assessment of donor-specific antibodies [9]. As soon as therapies such as immunoadsorption, plasmapheresis, or antibody treatment are carried out, a statement regarding the success of the therapy is useful and essential. To be able to make at least a statement on a relative scale, a ratio value can be determined from the MFI values if sera are tested in parallel before (same test performance on the same Luminex run) and after therapy using the same approach. There are several options to overcome this problem including a clear, statistically sound definition of a positive and negative reaction, including the respective true MFI value (tMFI) for any sample analyzed or the use of a ratio value for every mismatched bead using an earlier serum sample as a reference in a parallel assay. The latter would eliminate any day-to-day and user-to-user variation. Both options are shown below.

## 2. Materials and Methods

This analysis assessed variability in bead measurements from Luminex single-antigen bead (SAB) assays. A total of 358 raw MFI measurements were considered, derived from 61 patients across six different commercially available lots (LABScreen™ assay provided by One Lambda, West Hills, CA, USA) following the manufacturer’s recommendation.

The measurements span from 2017 to 2024. The cohort consists of 39 kidney, 19 liver, and 2 stem-cell-transplant patients and one patient who underwent treatment without a transplant (Appendix A).

The patients were selected to compare their MFI values pre- and post-transplantation or immunosuppressive therapy. The bead-based Luminex technology (Luminex 200^®^, Luminex Corporation, Austin, TX, USA) was used for HLA antibody testing, whereby the measurements were carried out with Xponent software version 4.3 and the HLA antibody analysis was performed as described by Lehmann et al. [10]. The bead coating for HLA class I and HLA class II is shown in Appendix A. For analysis, the HLA Fusion software version 4.4 was used. To provide a technical overview of the measurement variability, measurements with an MFI value above 1500 were filtered out. This resulted in approximately 20% of observations (16.50% in HLA class II, 18.95% in HLA class I) being removed.

By excluding these measurements, the analysis aims to focus on baseline variability across beads, reducing the impact of individual immunological responses or biases introduced by the specific cohort. Statistical analysis was performed on log10-transformed MFI values. Pairwise comparisons were performed using the Wilcoxon rank-sum test with Bonferroni correction for multiple testing if not stated otherwise. The adjusted *p*-values, effect sizes, and confidence intervals for groups showing significant standard deviation differences are listed in Appendix A.

For the potential donor-specific antibodies, a ratio was calculated to relate longitudinal MFI values to corresponding MFI values before treatment or transplantation (Figure 1). The latter refer to baseline serum MFI values, which are individually selected depending on the clinical purpose: for potential donor-specific antibody (DSA) monitoring after transplantation, the last serum sample taken before transplantation should be used, and for judging immune suppression efficacy, serum taken directly before the start of therapy should be used.

Importantly, baseline serum aliquots were measured in parallel to the respective current sample, such that the ratio was calculated from MFI values derived from the same Luminex run.

## 3. Results

### 3.1. Bead Variability (HLA Class I)

First, we are interested in the technical variability of the HLA-class-I-specific beads in a cohort of 61 individuals tested in the frame of clinical assessment preceding transplantation [10]. Therefore, we calculated the bead-wise standard deviation of the log10-transformed MFI values. Figure 2 shows this bead variability of HLA class I beads. We excluded all measurements above 1500 MFI to remove biological variability associated with antibody presence. This filter also removes virtually all positive control bead measurements. The HLA class I beads exhibited standard deviations (SDs) ranging from 0.41 to 0.63, with HLA-bead B*14:01 showing the lowest variability and A*11:01 the highest. A summary of all loci groups, their SDs, means, and the beads showing the lowest or highest SDs is provided in Table 1.

The three HLA loci in Figure 2 show a distinct pattern where HLA-C beads have the lowest SD (0.47) and HLA-A the highest SD (0.53), while all three groups are significantly different from each other (*p* < 0.05, Appendix A). These SDs correspond to a 2.95×–3.39×-fold change for linear MFI values. When the 1500 MFI filter is removed, the differences between HLA-A and HLA-B are no longer significant. The general pattern, however, still remains with the difference compared to HLA-C shifting even more (Appendix A, Appendix A).

Cross referencing the SDs with the mean values of the three groups in Table 1, we can see an inverse trend: HLA-C exhibits the highest mean value and lowest variability, and HLA-A shows the lowest mean and highest variability.

This pattern may be attributed to a higher baseline noise on HLA-C beads, giving it less range for fold changes. Consistent with this interpretation, analysis of the mean values of all self-alleles revealed that HLA-C loci displayed the highest MFI values and HLA-A the lowest (Figure 3). Significant differences in the mean self-allele MFI could only be found between HLA loci A and C in a Wilcoxon rank-sum test without Bonferroni correction (*p* < 0.05).

To determine whether these patterns were also displayed in individual lots, we examine the SDs from all lots that were considered for these analyses in Figure 4. The results suggest that variability is indeed lot-dependent. Locus SDs fluctuated across different lots with varying relations between them. HLA-B and HLA-C have a fairly constant relationship in all lots, with C having the lower and B the higher variation. The differences in these groups are also significant in lot 11–lot 14 (*p* < 0.05). Another noticeable trend is the variability of HLA-A compared to HLA-C, where A tends to be higher in lots 10–12 and 15 (*p* < 0.05). While most lots follow a slightly similar pattern, namely lot 13 and lot 14 show a different relation of the group SDs compared to the rest.

Lot-to-lot variability can also be observed when comparing the mean MFI values of the three groups (Appendix A). Here the only consistent relationship across all lots is between HLA-A and HLA-C, with group A showing a lower mean MFI value than HLA-C in every lot (*p* < 0.01), except for lot 12.

Overall, HLA class I beads display variability patterns among and within allele groups, with HLA-C mostly showing the lowest variability and highest baseline MFI values when observing self-allele measurements. Although this pattern generally persists across different lots, considerable lot-to-lot differences remain.

### 3.2. Bead Variability (HLA Class II)

Analyses for HLA-class-II-specific beads were performed analogously to HLA class I: First, we compared the variability between the loci, then we analyzed the antibody levels of the self-alleles, and finally, we compared variability across the lots. Figure 5 compares the variability for HLA class II beads. Here, SDs range from 0.38 to 0.62, which is comparable to class I variability (compare Table 1). The dual-coated bead HLA-DQA1*02:01–HLA-DQB1*02:02 shows the lowest and HLA-DRB1*16:02 the highest variability.

Specific patterns here were generally less noticeable. Significant differences between HLA class II loci could only be found between the DRB1 group and the dual-coated bead group DQA1/DQB1 (*p* < 0.05, Appendix A). When the 1500 MFI filter was removed, all groups showed increased SDs (Appendix A). The dual-coated beads DQA1/DQB1 showed the largest increase and became significantly different from the other dual-coated-bead group DPA1/DPB1 (*p* < 0.05, Appendix A).

Examining the mean values and SDs in Table 1, we did not observe the same inverse relationship between mean MFI and SD values as was seen in HLA class I. While the group with the highest SD (DRB1) does have the lowest mean, the groups with the lowest SDs (DPA1/DPB1 or DRB3) do not exhibit the highest means. This suggests that the relationship seen in HLA class I is not a general rule across all HLA bead types. Also, when observing self-allele measurements, we could not observe meaningful differences between the groups (Figure 6).

Generally, the range of MFI values appears to be more widespread, showing slight similarities with self HLA-C measurements (Figure 3). For dual-coated beads, we found a similar mean MFI when considering only measurements from patients who have both corresponding alleles (Figure 7).

Comparing the metrics shown in Figure 6 and Figure 7, we can see that the dual-coated beads have slightly lower MFI values than the single-coated beads with a lower variability. Overall, dual-coated beads show no major differences in SD when the MFI filter is applied (Figure 5) or when using self-allele raw MFI readings (Figure 6 and Figure 7). However, when observing the lot-to-lot variability*,* we can see that this relationship is not consistent over multiple lots (Figure 8). Only in lot 12 were significant differences between the two groups of DQA1/DQB1 and DPA1/DPB1 observed, with their relationship appearing to shift in the remaining lots. While lots 13 to 16 appear to have different variability patterns, none of the group SDs showed significant differences in a Wilcoxon rank-sum test with Bonferroni correction.

### 3.3. Median Fluorescence Intensity Ratio

Previously, we have presented a 2-year-old patient who received a living kidney transplant from his grandfather [10]. After 6 years, the patient received a second kidney transplant. HLA mismatch constellations at a high-resolution level between the recipient and donor as well as HLA antibody formation have already been described in detail in the corresponding article.

Following an episode of rejection after a second kidney transplant, various therapies, including plasmapheresis and antibody columns together with rituximab, were carried out to eliminate the donor-specific HLA antibodies formed de novo in the patient. To be able to assess the success of the therapy, the change in the anti-HLA antibody profile should be taken into account: Therefore, an early serum sample of the patient was determined as the reference serum, and all measurements thereafter were performed using the current serum and an aliquot of the reference in parallel. Then, the respective ratios were calculated for all bead MFIs.

In Figure 9, the MFI and the derived ratios are shown through the course of 16 months following organ transplantation. Paradigmatically, donor-specific antibodies of three HLA mismatches are shown: HLA-A24 (Figure 9a), HLA-DR52 (Figure 9b), and HLA-DQ7 (Figure 9c). Of note is that the diagrams represent the mean MFI value averaged over all beads of the antigen group.

For HLA-A24, both the ratio and MFI value are almost identical and follow the same pattern: we see a steep decay in the diagrams following immunosuppression, followed by relatively steady antibody levels. This is different for the antigen groups HLA-DR52 and HLA-DQ7: the MFI curve is highly variable, ranging from values of around 1500 to more than 15,000 (orange line in Figure 9b,c). The ratio, in contrast, shows less variability (blue line).

The second transplantation occurred on 9 June 2022. Between this date and 11 July 2022, the patient suffered from an antibody-mediated acute rejection treated by rituximab and six plasmaphereses, resulting in a drop in the ratio for the HLA-class-I-specific antibodies to below 1. Virtually no changes were observed for the observation period depicted here. With regard to the HLA-class-II-specific antibodies, the situation differs, as shown in Figure 9b, c. Both the MFI values and ratios seem to correlate in the first post-transplant phase. This changes shortly thereafter. In the period between 6 October 2022 and 17 November 2022, the MFI value continues to drop towards the cutoff, while the ratio shows an increase, suggesting a rejection episode. This was seen for both the HLA-DR52 and the HLA-DQ7 DSAs. The patient suffered from a transplant infection and a reduced function of the organ. An opposite observation was seen between 7 March 2023 and 25 April 2023, where the MFI increased but the ratios remained at low levels. The patient suffered from a severe COVID-19 infection. The transplant remained stable until 24 January 2025, the last date of observation in the present report. The results may be interpreted as follows: shifts in the ratios reflect events within the transplanted organ, whereas changes in the MFI values indicate opportunistic infections. To establish this approach as a diagnostic framework, these observations must be validated in a larger cohort.

These results show that the utilization of ratios is less sensitive with regard to technical bias, especially when considering the variability described above. This robustness, however, comes at the cost of measuring two sera (current + reference) each time.

## 4. Discussion

The current report shows that the data obtained after Luminex single-antigen bead assay testing for HLA-specific antibodies are of relative value [11,12]. Using the crude data for quantitative statements or antibody concentration, or a decision-making procedure, i.e., adjustment of immunosuppressive treatment or other supportive therapy, is premature and needs clarification [13].

The analysis of the HLA class I bead results shows variability in the day-to-day, lot-to-lot, user-to-user and laboratory-to-laboratory [5] results. Interestingly, even the locus-to-locus results show a significant variation. The SDs range from 0.44 to 0.50 in log10 space, which corresponds to an approximate three-fold span in retransformed linear space. In other words, a repeat measurement can be up to three times higher or one third lower in the next measurement. Lifting the MFI filter only increases this variability (Appendix A).

The analysis of HLA class II beads revealed less distinct variability patterns compared to class I, though significant differences were observed between DRB1 and the dual-coated groups of DQA1/DQB1 and DPA1/DPB1. Lifting the 1500 MFI filter from this analysis resulted in all groups having larger means and SDs, as expected. The DQA1/DQB1 group’s SD in particular increased drastically, and this resulted in it being significantly different from that of DPA1/DPB1 (*p* < 0.05, Appendix A and Appendix A).

Unlike class I, class II did not exhibit an inverse relationship between the mean MFI and SD, suggesting that not all variability can be attributed to baseline noise or self-allele measurements. Notably, the analysis of self-allele beads revealed many cases with clearly measurable MFI values, particularly among HLA-C beads. This observation appears to be characteristic of One Lambda kits, as it has also been reported by Karahan et al. [14].

Given the lot-to-lot variability in HLA classes I and II (Figure 4 and Figure 8), batch effects remain a critical factor when interpreting Luminex results.

In our opinion, taking any treatment-modification decisions using these data seems quite unreliable. We tried to circumvent the problem by calculating a ratio between the current serum and a control serum obtained before transplantation, as also suggested by Sullivan et al. [6]. This is shown paradigmatically for HLA class I and II mismatches. The determination of a ratio value has several advantages, such as the personalized evaluation of MFI data and the possibility of individual statements regarding DSAs. The analysis with ratios of >1, =1, or <1 provides a simple semi-quantitative tool for the personalized monitoring of a patient’s therapy and evaluation of its effectiveness. Of note, we have focused here on the comparability of longitudinal measurements of one person during follow-up antibody monitoring or immunosuppressive therapy, and not on user-to-user comparability.

Not every MFI value increase is a real increase in the antibody presence, as can be seen in Figure 9b,c. However, this procedure implies additional costly testing, and the selection of a representative, meaningful comparative serum is essential. The choice of serum should reflect the specific medical question and be made in close consultation with the treating clinician. The reference may also change over time as the clinical context and treatment evolve. Lastly it is recommended that a reference serum will be handled in aliquots, as freeze–thaw cycles might have an impact on the antibodies in the serum.

Given these challenges, it may be more practical to implement a method that allows for recalculation of the MFI using a standard [9], which allows the calculation of HLA class I and HLA class II International Units, as has been carried out for other immunological tests [15,16,17]. However, there are no globally standardized controls for anti-HLA antibodies, so MFI changes cannot be analyzed quantitatively.

The degree of variability within repeated measurements of the same sample raises concerns of using a fixed MFI threshold to define antibody presence. It also shows that reporting MFI values alone and using them for statistical interpretation without context can lead to misinterpretation. While running a reference serum does mitigate some of the variability and allows for more meaningful interpretation, it is costly and comes with its own set of challenges. To combat this, we opt to investigate alternative methods of MFI value interpretation, e.g., by using dynamic MFI cutoffs.

Claas [9] suggested the following in 2010: “The clinical relevance of the presence of donor-specific HLA antibodies before transplantation and the appearance of these antibodies after transplantation is a controversial issue. The lack of consensus between different centers is partly due to the heterogeneity of the HLA antibodies involved. Standardization is essential and future studies must focus on the further characterization of the antibody titers, the immunoglobulin (sub)classes of the antibodies and the epitopes recognized. It remains to be established in which cases the HLA antibodies are the direct cause of or just associated with (chronic) graft failure”.

## 5. Conclusions

Our study shows that inherent variability across HLA class I and HLA class II systems must be considered when comparing MFI values between alleles, loci, or individuals. For longitudinal antibody monitoring of one individual over time, we demonstrated a ratio-based approach to transform MFI values into comparable changes with respect to a baseline measurement.

## Figures and Tables

**Figure 1 biology-14-00686-f001:**
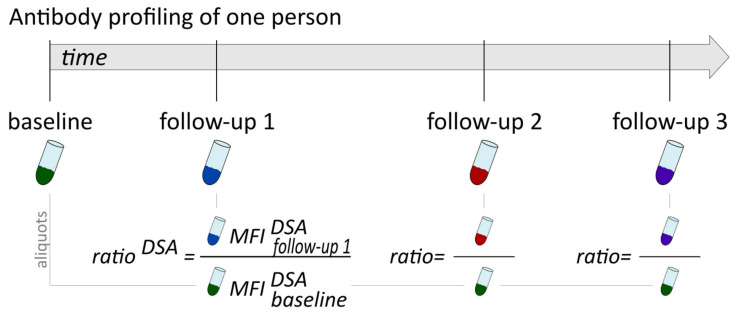
Longitudinal antibody profiling using ratios. A baseline sample (e.g., pre-treatment or pre-transplantation) is used as reference for calculation of antibody change rates in follow-up measurements. Therefore, antibody-specific ratios are calculated as the quotient of the follow-up MFI and the corresponding baseline MFI.

**Figure 2 biology-14-00686-f002:**
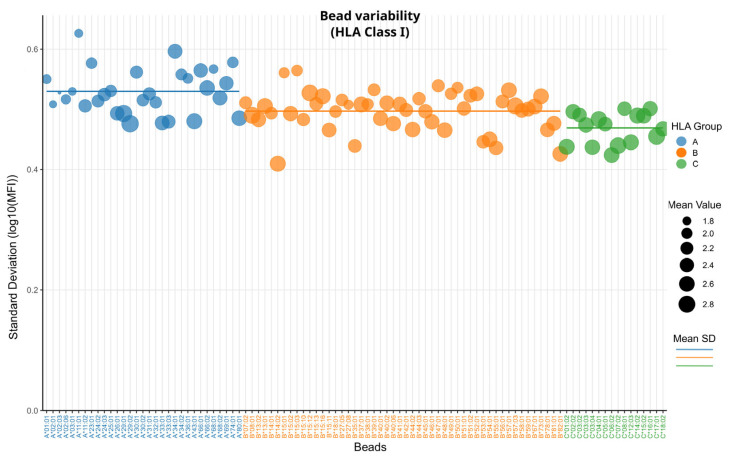
HLA class I bead variability: The Y-axis shows the standard deviation of log10-transformed MFI values of each bead. Each dot represents a bead, with its size proportional to the mean MFI value of all measurements. Dots are colored according to HLA locus; horizontal lines represent the mean SD for each locus.

**Figure 3 biology-14-00686-f003:**
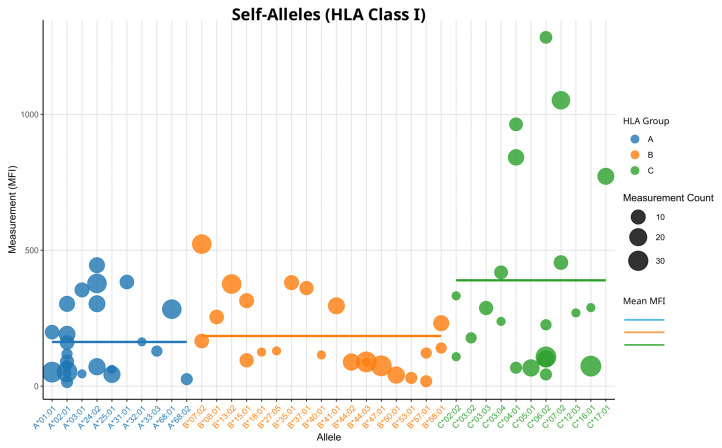
Measurements of all self-alleles for HLA class I beads: The Y-axis shows raw MFI values. Every bead represents the measurements of one patient. If there is more than one measurement, the average is used. Size of the dots is proportional to the number of multiple measurements. Dots are colored according to HLA locus, horizontal lines represent the mean MFI for each HLA locus, respectively.

**Figure 4 biology-14-00686-f004:**
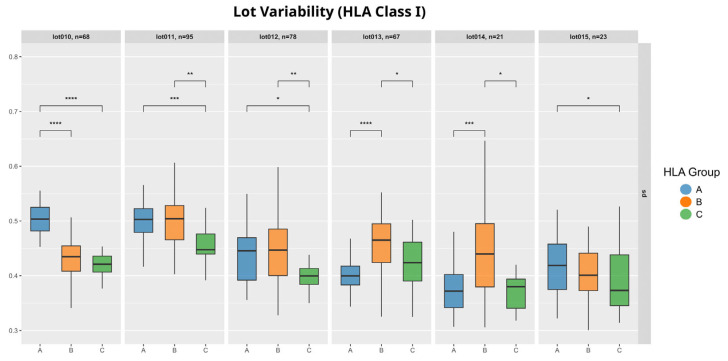
Variability analysis by lot: Each box represents standard deviation in log10 of one HLA class I locus, and the loci are grouped by LABScreen single-antigen lot. Significant differences between the HLA loci were identified using Wilcoxon rank-sum test without Bonferroni correction (significance codes: **** *p* < 0.0001, *** *p* < 0.001, ** *p* < 0.01, * *p* < 0.05).

**Figure 5 biology-14-00686-f005:**
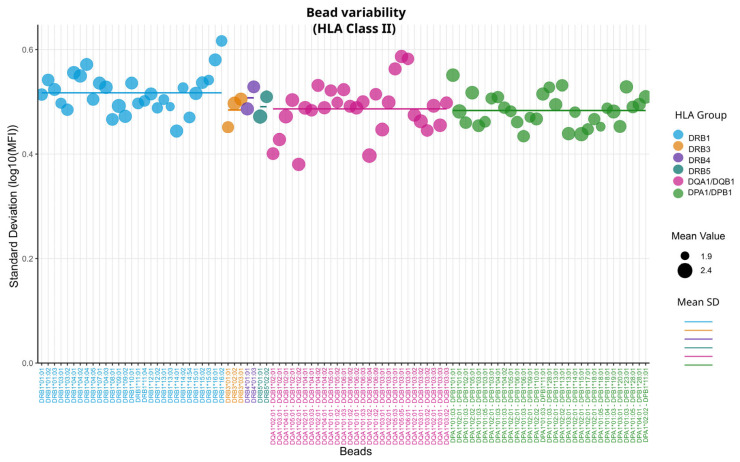
Bead variability of HLA class II. The Y-axis shows the standard deviation of log10-trans-formed MFI values of each bead. Each dot represents a bead, with its size proportional to the mean MFI value of all measurements. Dots are colored according to HLA locus; horizontal lines represent the mean SD for each locus.

**Figure 6 biology-14-00686-f006:**
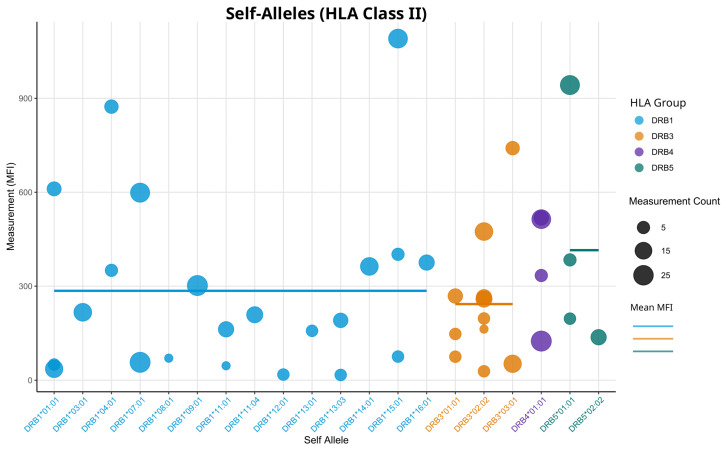
Measurements of all self-alleles for HLA class II single-coated beads. The Y-axis shows raw MFI values. Every bead represents the measurements of one patient. If there is more than one measurement, the average is used. Size of the dots is proportional to the number of multiple measurements. Dots are colored according to HLA locus; horizontal lines represent the mean MFI for each locus.

**Figure 7 biology-14-00686-f007:**
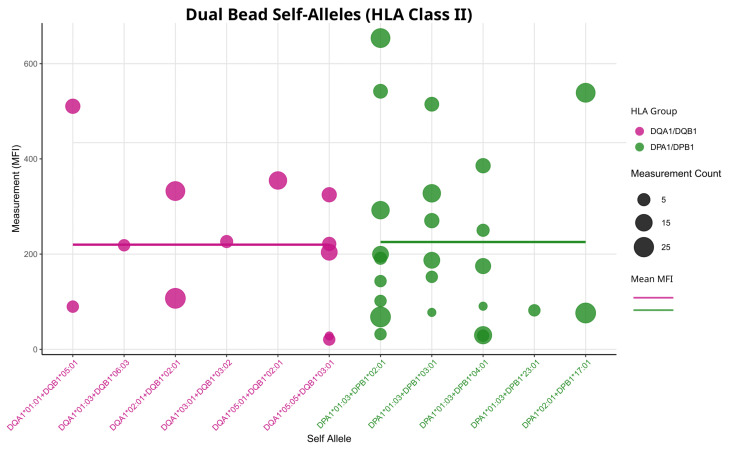
Measurements of all self-alleles for class II dual-coated beads. The Y-axis shows raw MFI values. Every bead represents the measurements of one patient. If there is more than one measurement, the average is used. Size of the dots is proportional to the number of multiple measurements. Dots are colored according to HLA locus; horizontal lines represent the mean MFI for each locus.

**Figure 8 biology-14-00686-f008:**
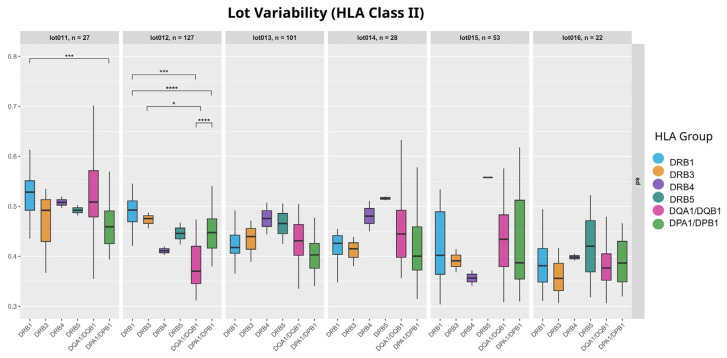
Lot variability for HLA class II. Each box represents SD of one locus, and the loci are grouped by Luminex lot. Significant differences between the loci were identified using Wilcoxon rank-sum test with Bonferroni correction (significance codes: **** *p* < 0.0001, *** *p* < 0.001, * *p* < 0.05).

**Figure 9 biology-14-00686-f009:**
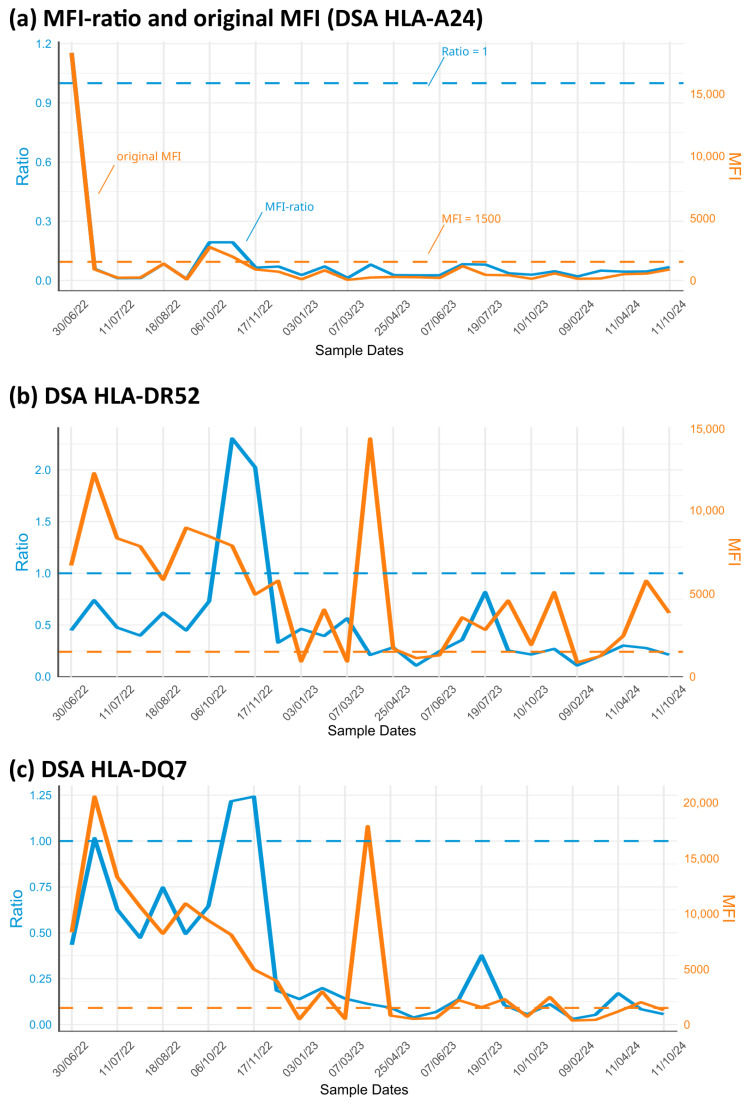
Comparison between ratio (blue curve) and the respective MFI values (orange curve) obtained from sera of the patient after transplantation: Three HLA mismatches were selected: (**a**) indicates donor-specific antibodies (DSAs) of HLA-A24; (**b**) of HLA-DR52, and (**c**) of HLA-DQ7.

**Table 1 biology-14-00686-t001:** Variability of HLA class I and HLA class II beads. Mean MFI and corresponding SD were calculated on log10-transformed MFI values. Fold change is calculated by retransforming the log10 SD into linear space ($10^{SD}$). Beads with the highest and lowest SDs are given for each HLA locus.

HLA Locus	Mean	SD	Fold Change	Max SD	Min SD
A	2.09	0.53	3.38	A*11:01 (0.63)	A*29:02 (0.48)
B	2.18	0.50	3.16	B*15:03 (0.56)	B*14:02 (0.41)
C	2.26	0.47	2.95	C*16:01 (0.50)	C*06:02 (0.42)
DPA1/DPB1	2.11	0.48	3.02	DPA1*01:03–DPB1*01:01 (0.55)	DPA1*01:03–DPB1*06:01 (0.43)
DQA1/DQB1	2.17	0.49	3.10	DQA1*05:05–DQB1*03:01 (0.59)	DQA1*02:01–DQB1*02:02 (0.38)
DRB1	2.08	0.52	3.31	DRB1*16:02 (0.62)	DRB1*14:01 (0.44)
DRB3	2.15	0.48	3.02	DRB3*03:01 (0.51)	DRB3*01:01 (0.45)
DRB4	2.14	0.51	3.24	DRB4*01:03 (0.53)	DRB4*01:01 (0.49)
DRB5	2.20	0.49	3.10	DRB5*02:02 (0.51)	DRB5*01:01 (0.47)

## Data Availability

The data supporting this study’s findings are available from the corresponding authors upon reasonable request.

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
