# Peer review of "Luminex MFI—Efforts from a Qualitative to a Quantitative Analysis"

_biology, 2025, doi:10.3390/biology14060686_

Round 1
Reviewer 1 Report
Comments and Suggestions for Authors
This manuscript deals with Luminex-MFI and its clinical significance. The article is very well written and deals with a very relevant topic in transplantation immunology. Therefore, after some minor changes, the article should be published.
Comment 1: Page 2, line 60
The authors refer here to 6 different reagent lots. Which ones were used here, commercially available or in-house lots? Please elaborate on this.
Comment 2: Figure 2
Regarding the measurement of self-alleles, why are MFI values clearly measurable here? Almost no MFI values are detected in commercial kits. Please comment on this.
Comment 3: Page 1, line 25-26
The term "ratio" is briefly described here. This should be explained in more detail in the introduction for a better understanding for the reader. In addition, the difference to the normalized background (NBG) ratio should be described.
Comment 4: Page 8, line 215
Please correct “plasmaphereses” to “plasmapheresis”.
Comment 5: Page 11, line 305
Please remove the following character “.
Comment 6: Page 11, line 270
You mention a “user to user” comparison here. It would be desirable to describe the differences between the individual users here as well.
Author Response
Thank you for your efforts and your positive review of our manuscript. We addressed all points raised and revised the manuscript accordingly. Below you find detailed replies.
Thank you very much
Comment 1: The authors refer here to 6 different reagent lots.
Answer: We only use commercial test kits with CE certification according to EFI accreditation of our laboratory. We added this information to the material & Methods section.
Comment 2: Regarding the measurement of self-alleles, why are MFI values clearly measurable here?
Answer: Thank you for hinting at this important point. This is a situation with OneLambda tests (esp. HLA-C) as previously reported (Karahan et al.: Comparison of different luminex single antigen bead kits for memory B cell-derived HLA antibody detection. HLA 2021). We added this to the discussion section.
Comment 3: The term "ratio" is briefly described here. This should be explained in more detail in the introduction.
Answer: We agree that the description was not clear enough. Therefore, we added a schematic as new Figure 1 in the M&M section, which explains how we calculate the ratio. We also revised and extended the corresponding description. We also state that we concentrate on detectable and potential DSA when using this approach.
Comments 4&5: Thank you, we corrected these typos.
Comment 6: You mention a “user to user” comparison here. It would be desirable to describe the differences between the individual users here as well.
Answer: The user-to-user comparison was not the focus of our study, we here focus on comparability of longitudinal measurements of one person during follow-up or therapy. We highlighted this in M&M and discussion sections.
Reviewer 2 Report
Comments and Suggestions for Authors
The manuscript presented for review titled “Luminex-MFI – efforts from a qualitative to a quantitative analysis” presents an insightful evaluation of the technical limitations associated with the use of mean fluorescence intensity (MFI) in Luminex single-antigen bead (SAB) assays for HLA antibody detection. The authors introduce a new semi-quantitative metric called the MFI ratio (comparing post-treatment to pre-treatment levels), and back it up with strong empirical evidence drawn from both a patient cohort and a detailed clinical case study.
The subject matter of this article is important for transplant immunology, given the continued reliance on semi-quantitative data for critical therapeutic decisions.
However, certain areas of the manuscript require improvement and clarification before it is suitable for publication.
- The manuscript offers a strong justification for employing MFI ratios to mitigate intra-assay and lot-to-lot variability in Luminex-based HLA antibody testing. It is important to recognize that the method remains semi-quantitative and is not calibrated against globally accepted reference standards, such as the WHO International Standard for anti-HLA antibodies. As a result, this limits the method’s generalizability and cross-laboratory reproducibility. I recommend that the authors explicitly acknowledge this limitation in the discussion section. Authors should include a brief commentary on how the absence of absolute calibration limits comparability across laboratories and institutions.
- In the methodology, the authors exclude all MFI values greater than 1500 to focus on assay-related variability and reduce biological signal. While the rationale to reduce immunological noise is understandable, this exclusion may introduce unintended selection bias. Specifically, it omits high-affinity or strongly reactive HLA antibodies, which are often the most clinically relevant. This filtering could limit the generalizability of the variability analysis, especially when assessing the robustness of the ratio method in contexts where strong donor-specific antibodies are present. I recommend that the authors include a sensitivity analysis to illustrate variability patterns with and without the >1500 MFI cutoff. Alternatively, stratifying the analysis by MFI intensity ranges (e.g., <1000, 1000–5000, >5000) could provide valuable insights into the method’s performance across the full biological spectrum. At a minimum, this limitation should be explicitly addressed in the discussion, particularly with respect to its implications for applying the method in contexts involving high antibody levels.
- The current manuscript demonstrates the utility of the MFI ratio metric primarily through a single case study, which limits the generalizability of the findings. To strengthen the validity of the proposed approach, authors should consider the inclusion of a small validation cohort—such as 5 to 10 transplant recipients with well-characterized clinical outcomes—would be highly beneficial and significantly enhance confidence in the method’s broader applicability.
- Although the authors rightly employ a log10 transformation to address the skewness of MFI distributions, the manuscript lacks clarity on the clinical interpretation of variability in the transformed scale. For example, a standard deviation of 0.5 in log10 space reflects approximately a 3.16-fold difference in raw MFI values—an extent of variation that could significantly impact the identification of clinically relevant donor-specific antibodies. Providing a clearer translation between statistical variation and clinical thresholds would greatly improve the method’s interpretability and practical relevance.
- The manuscript presents p-values from various statistical comparisons but does not include adjusted p-values, effect sizes, or confidence intervals. Incorporating these additional metrics would significantly improve statistical transparency and enable readers to more accurately interpret the magnitude and reliability of the observed effects, particularly in the context of multiple group comparisons.
Author Response
Thank you for your efforts and your positive review of our manuscript. We addressed all points raised and revised the manuscript accordingly. Below you find detailed replies.
Thank you very much
Comment 1: It is important to recognize that the method remains semi-quantitative and is not calibrated against globally accepted reference standards, such as the WHO International Standard for anti-HLA antibodies. As a result, this limits the method’s generalizability and cross-laboratory reproducibility. I recommend that the authors explicitly acknowledge this limitation in the discussion section.
Answer 1: There are no globally standardized controls for anti-HLA antibody testing via SAB technique, so MFI changes cannot be analyzed quantitatively. Such a standard exists, for example, for SARS-Cov2 antibody tests (WHO standard). In the HLA laboratories, controls are often generated in house, from immunized patient sera. Quantitative comparability between laboratories is not possible here. There is a negative control serum from the kit manufacturer, but there are no standardized positive controls with a specific unit available. We indicate this in the discussion section.
Further, the aim of our study was not an inter-laboratory comparison. We here focus on comparability of longitudinal measurements of one person during follow-up or therapy. We highlighted this in M&M and discussion sections.
Comment 2: In the methodology, the authors exclude all MFI values greater than 1500 to focus on assay-related variability and reduce biological signal. While the rationale to reduce immunological noise is understandable, this exclusion may introduce unintended selection bias.
Answer 2: This is absolutely correct. Removing this filter step even increases variability (see below) and therefore also increases incomparability, which is a main message of our manuscript. To further investigate an unintended effect of the filter, we performed variability analyses without removal of measurements with MFI>1,500: Lifting the 1,500 MFI filter from this analysis expectedly resulted in all groups having larger means and SDs, as expected. The general pattern however still remains virtually unchanged. We added these findings in the Results and Discussion sections, and in the new Figures S1 & S3 and Table S5.
Comment 3: To strengthen the validity of the proposed approach, authors should consider the inclusion of a small validation cohort.
Answer 3: In this study, only one case is presented and discussed as an example of ratio determination. We have a total of more than 40 patients in whom the ratio was determined as part of therapy success determination or long-term follow-up after organ transplantation. These data, together with clinical data, are currently being analyzed and will be presented separately.
Comment 4: Although the authors rightly employ a log10 transformation to address the skewness of MFI distributions, the manuscript lacks clarity on the clinical interpretation of variability in the transformed scale. … Providing a clearer translation between statistical variation and clinical thresholds would greatly improve the method’s interpretability and practical relevance.
Answer 4: We agree that data transformation into log-space is not intuitive and somewhat unusual especially for lab-orientated or clinical readers. Therefore, we now provide a re-transformed ‘fold change’ as equivalent of the changes in linear MFI space in Table 1 to convey a feeling for these values.
Comment 5: The manuscript presents p-values from various statistical comparisons but does not include adjusted p-values, effect sizes, or confidence intervals.
Answer 5: Thank you for this suggestion. We accordingly added Table S4, which provides adjusted p values, effect sizes and confidence intervals for locus comparisons.
Reviewer 3 Report
Comments and Suggestions for Authors
This paper investigates variability in Luminex Single Antigen Bead (SAB) assay measurements for HLA antibody testing in transplantation. The authors examine technical variability across different HLA loci, lots, and measurement conditions, ultimately proposing a ratio-based approach to address the inherent limitations of mean fluorescence intensity (MFI) values. I have several comments as follows:
- The authors acknowledge the limitations of current approaches but don't provide detailed implementation guidance for their ratio method. Questions remain about how to select appropriate baseline samples, what ratio thresholds should trigger clinical action, and how to standardize this approach across centers.
- Figure 3 and Figure 7 has missing legend for the color, though I think the x-axis means the same thing? It's not very clear.
-
Figure 4: Bead variability of HLA class II analogue to Figures 1. It should be Figure 1. There are other typos like this in the manuscript too and I suggest authors to fix them all.
- The analysis includes 358 measurements from 59 kidney transplant patients, but there's insufficient information about patient demographics, clinical characteristics, or why these specific patients were selected. This limits the generalizability of the findings.
Author Response
Thank you for your efforts and your positive review of our manuscript. We addressed all points raised and revised the manuscript accordingly. Below you find detailed replies.
Thank you very much
Comment 1: The authors acknowledge the limitations of current approaches but don't provide detailed implementation guidance for their ratio method.
Answer 1: We agree that the description was not clear enough. Therefore, we added a schematic as new Figure 1 in the M&M section, which explains how we calculate the ratio. We also revised and extended the corresponding description.
Comment 2: Questions remain about how to select appropriate baseline samples, what ratio thresholds should trigger clinical action, and how to standardize this approach across centers.
Answers 2: Indeed, selection of the baseline serum is crucial and depends on clinical purpose: For potential donor specific antibodies (DSA) monitoring after transplantation, the last serum before transplantation should be used, and for judging immune suppression efficacy a serum taken directly before therapy start. We added this to the Material&Methods section, and discuss it in the discussion section.
In this study, only one case is presented and discussed as an example of ratio determination. We have a total of more than 40 patients in whom the ratio was determined as part of therapy success determination or long-term follow-up after organ transplantation. These data, together with clinical data, are currently being analyzed and will be presented separately.
A user-to-user comparison or comparability across centers was not the focus of our study, we here focus on comparability of longitudinal measurements of one person during follow-up or therapy. We highlighted and discussed this in M&M and discussion sections.
Comment 3: Figure 3 and Figure 7 has missing legend for the color.
Answer 3: Thank you for this feedback, we added color legends to these figures.
Comment 4: There are other typos like this in the manuscript too and I suggest authors to fix them all.
Answer4: We apologize for these mistakes. We revised the manuscript and corrected several typos.
Comment 5: The analysis includes 358 measurements from 59 kidney transplant patients, but there's insufficient information about patients.
Answer 5: We agree that additional information of the patients is valuable for this study, we therefore created an additional table (Table S1) with corresponding cohort information.
Round 2
Reviewer 2 Report
Comments and Suggestions for Authors
I would like to thank the authors for their detailed and thoughtful replies to all comments. I am satisfied with the revisions and the clarifications provided. All concerns have been adequately addressed, and I have no further comments at this time.